# A Composable Specification Language for Reinforcement Learning Tasks

**Kishor Jothimurugan, Rajeev Alur, Osbert Bastani**
University of Pennsylvania
{kishor,alur,obastani}@cis.upenn.edu

## Abstract

Reinforcement learning is a promising approach for learning control policies for robot tasks. However, specifying complex tasks (e.g., with multiple objectives and safety constraints) can be challenging, since the user must design a reward function that encodes the entire task. Furthermore, the user often needs to manually shape the reward to ensure convergence of the learning algorithm. We propose a language for specifying complex control tasks, along with an algorithm that compiles specifications in our language into a reward function and automatically performs reward shaping. We implement our approach in a tool called SPECTRL, and show that it outperforms several state-of-the-art baselines.

## 1 Introduction

Reinforcement learning (RL) is a promising approach to learning control policies for robotics tasks [5, 21, 16, 15]. A key shortcoming of RL is that the user must manually encode the task as a real-valued reward function, which can be challenging for several reasons. First, for complex tasks with multiple objectives and constraints, the user must manually devise a single reward function that balances different parts of the task. Second, the state space must often be extended to encode the reward—e.g., adding indicators that keep track of which subtasks have been completed. Third, oftentimes, different reward functions can encode the same task, and the choice of reward function can have a large impact on the convergence of the RL algorithm. Thus, users must manually design rewards that assign "partial credit" for achieving intermediate goals, known as *reward shaping* [17].

For example, consider the task in Figure 1, where the state is the robot position and its remaining fuel, the action is a (bounded) robot velocity, and the task is

> "Reach target $q$, then reach target $p$, while maintaining positive fuel and avoiding obstacle $O$".

To encode this task, we would have to combine rewards for (i) reaching $q$, and then reaching $p$ (where "reach x" denotes the task of reaching an $\epsilon$ box around $x$—the regions corresponding to $p$ and $q$ are denoted by $P$ and $Q$ respectively), (ii) avoiding region $O$, and (iii) maintaining positive fuel, into a single reward function. Furthermore, we would have to extend the state space to keep track of whether $q$ has been reached—otherwise, the control policy would not know whether the current goal is to move towards $q$ or $p$. Finally, we might need to shape the reward to assign partial credit for getting closer to $q$, or for reaching $q$ without reaching $p$.

We propose a language for users to specify control tasks. Our language allows the user to specify objectives and safety constraints as logical predicates over states, and then compose these primitives

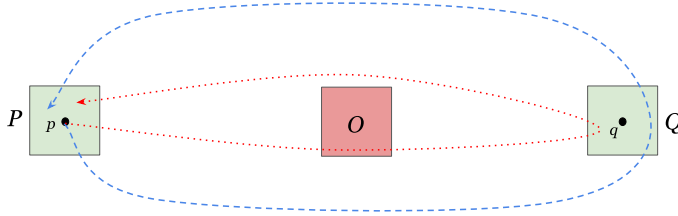

Figure 1: Example control task. The blue dashed trajectory satisfies the specification $\phi_{\text{ex}}$ (ignoring the fuel budget), whereas the red dotted trajectory does not satisfy $\phi_{\text{ex}}$ as it passes through the obstacle.

sequentially or as disjunctions. For example, the above task can be expressed as

$$\phi_{\text{ex}} = \texttt{achieve (reach } q; \texttt{ reach } p) \texttt{ ensuring (avoid } O \wedge \texttt{fuel} > 0), \qquad (1)$$

where `fuel` is the component of the state space keeping track of how much fuel is remaining.

The principle underlying our approach is that in many applications, users have in mind a sequence of high-level actions that are needed to accomplish a given task. For example, $\phi_{\text{ex}}$ may encode the scenario where the user wants a quadcopter to fly to a location $q$, take a photograph, and then return back to its owner at position $p$, while avoiding a building $O$ and without running out of battery. Alternatively, a user may want to program a warehouse robot to go to the next room, pick up a box, and then bring this item back to the first room. In addition to specifying sequences of tasks, users can also specify choices between multiple tasks (e.g., bring back any box).

Another key aspect of our approach is to allow the user to specify a task without providing the low-level sequence of actions needed to accomplish the task. Instead, analogous to how a compiler generates machine code from a program written by the user, we propose a compiler for our language that takes the user-provided task specification and generates a control policy that achieves the task. RL is a perfect tool for doing so—in particular, our algorithm compiles the task specification to a reward function, and then uses state-of-the-art RL algorithms to learn a control policy. Overall, the user provides the high-level task structure, and the RL algorithm fills in the low-level details.

A key challenge is that our specifications may encode rewards that are not Markov—e.g., in $\phi_{\text{ex}}$, the robot needs memory that keeps track of whether its current goal is `reach` $q$ or `reach` $p$. Thus, our compiler automatically extends the state space using a *task monitor*, which is an automaton that keeps track of which subtasks have been completed.[1] Furthermore, this automaton may have nondeterministic transitions; thus, our compiler also extends the action space with actions for choosing state transitions. Intuitively, there may be multiple points in time at which a subtask is considered completed, and the robot must choose which one to use.

Another challenge is that the naïve choice of rewards—i.e., reward 1 if the task is completed and 0 otherwise—can be very sparse, especially for complex tasks. Thus, our compiler automatically performs two kinds of reward shaping based on the structure of the specification—it assigns partial credit for (i) partially accomplishing intermediate subtasks, and (ii) for completing more subtasks. For deterministic MDPs, our reward shaping is guaranteed to preserve the optimal policy; we empirically find it also works well for stochastic MDPs.

We have implemented our approach in a tool called SPECTRL,[2] and evaluated the performance of SPECTRL compared to a number of baselines. We show that SPECTRL learns policies that solve each task in our benchmark with a success rate of at least 97%. In summary, our contributions are:

- We propose a language for users to specify RL tasks (Section 2).
- We design an algorithm for compiling a specification into an RL problem, which can be solved using standard RL algorithms (Section 3).
- We have implemented SPECTRL, and empirically demonstrated its benefits (Section 4).

**Related work.** Imitation learning enables users to specify tasks by providing *demonstrations* of the desired task [18, 1, 26, 20, 10]. However, in many settings, it may be easier for the user to directly specify the task—e.g., when programming a warehouse robot, it may be easier to specify waypoints describing paths the robot should take than to manually drive the robot to obtain demonstrations.

Also, unlike imitation learning, our language allows the user to specify global safety constraints on the robot. Indeed, we believe our approach complements imitation learning, since the user can specify some parts of the task in our language and others using demonstrations.

Another approach is for the user to provide a *policy sketch*—i.e., a string of tokens specifying a sequence of subtasks [2]. However, tokens have no meaning, except equal tokens represent the same task. Thus, policy sketches cannot be compiled to a reward function, which must be provided separately.

Our specification language is based on *temporal logic* [19], a language of logical formulas for specifying constraints over (typically, infinite) sequences of events happening over time. For example, temporal logic allows the user to specify that a logical predicate must be satisfied at some point in time (e.g., "eventually reach state $q$") or that it must always be satisfied (e.g., "always avoid an obstacle"). In our language, these notions are represented using the `achieve` and `ensuring` operators, respectively. Our language restricts temporal logic in a way that enables us to perform reward shaping, and also adds useful operators such as sequencing that allow the user to easily express complex control tasks.

Algorithms have been designed for automatically synthesizing a control policy that satisfies a given temporal logic formula; see [4] for a recent survey, and [12, 25, 6, 9] for applications to robotic motion planning. However, these algorithms are typically based on exhaustive search over control policies. Thus, as with finite-state planning algorithms such as value iteration [22], they cannot be applied to tasks with continuous state and action spaces that can be solved using RL.

Reward machines have been proposed as a high-level way to specify tasks [11]. In their work, the user provides a specification in the form of a finite state machine along with reward functions for each state. Then, they propose an algorithm for learning multiple tasks simultaneously by applying the Q-learning updates across different specifications. At a high level, these reward machines are similar to the task monitors defined in our work. However, we differ from their approach in two ways. First, in contrast to their work, the user only needs to provide a high-level logical specification; we automatically generate a task monitor from this specification. Second, our notion of task monitor has a finite set of registers that can store real values; in contrast, their finite state reward machines cannot store quantitative information.

The most closely related work is [13], which proposes a variant of temporal logic called *truncated LTL*, along with an algorithm for compiling a specification written in this language to a reward function that can be optimized using RL. However, they do not use any analog of the task monitor, which we demonstrate is needed to handle non-Markovian specifications. Finally, [24] allows the user to separately specify objectives and safety constraints, and then using RL to learn a policy. However, they do not provide any way to compose rewards, and do not perform any reward shaping. Also, their approach is tied to a specific RL algorithm. We show empirically that our approach substantially outperforms both these approaches.

Finally, an alternative approach is to manually specify rewards for sub-goals to improve performance. However, many challenges arise when implementing sub-goal based rewards—e.g., how does achieving a sub-goal count compared to violating a constraint, how to handle sub-goals that can be achieved in multiple ways, how to ensure the agent does not repeatedly obtain a reward for a previously completed sub-goal, etc. As tasks become more complex and deeply nested, manually specifying rewards for sub-goals becomes very challenging. Our system is designed to automatically solve these issues.

## 2 Task Specification Language

**Markov decision processes.** A *Markov decision process (MDP)* is a tuple $(S, D, A, P, T)$, where $S \subseteq \mathbb{R}^n$ are the states, $D$ is the initial state distribution, $A \subseteq \mathbb{R}^m$ are the actions, $P : S \times A \times S \to [0, 1]$ are the transition probabilities, and $T \in \mathbb{N}$ is the time horizon. A *rollout* $\zeta \in Z$ of length $t$ is a sequence $\zeta = s_0 \xrightarrow{a_0} \ldots \xrightarrow{a_{t-1}} s_t$ where $s_i \in S$ and $a_i \in A$. Given a (deterministic) *policy* $\pi : Z \to A$, we can generate a rollout using $a_i = \pi(\zeta_{0:i})$. Optionally, an MDP can also include a reward function $R : Z \to \mathbb{R}$. [3]

**Specification language.** Intuitively, a specification $\phi$ in our language is a logical formula specifying whether a given rollout $\zeta$ successfully accomplishes the desired task—in particular, it can be interpreted as a function $\phi : Z \to \mathbb{B}$, where $\mathbb{B} = \{\texttt{true}, \texttt{false}\}$, defined by

$$\phi(\zeta) = \mathbb{I}[\zeta \text{ successfully achieves the task}],$$

where $\mathbb{I}$ is the indicator function. Formally, the user first defines a set of *atomic predicates* $P_0$, where every $p \in P_0$ is associated with a function $[\![p]\!] : S \to \mathbb{B}$ such that $[\![p]\!](s)$ indicates whether $s$ satisfies $p$. For example, given $x \in S$, the atomic predicate

$$[\![\texttt{reach } x]\!](s) \;=\; (\|s - x\|_\infty < 1)$$

indicates whether the robot is in a state near $x$, and given a rectangular region $O \subseteq S$, the atomic predicate

$$[\![\texttt{avoid } O]\!](s) \;=\; (s \notin O)$$

indicates if the robot is avoiding $O$. In general, the user can define a new atomic predicate as an arbitrary function $[\![p]\!] : S \to \mathbb{B}$. Next, *predicates* $b \in \mathcal{P}$ are conjunctions and disjunctions of atomic predicates. In particular, the syntax of predicates is given by [4]

$$b \;::=\; p \mid (b_1 \wedge b_2) \mid (b_1 \vee b_2),$$

where $p \in \mathcal{P}_0$. Similar to atomic predicates, each predicate $b \in \mathcal{P}$ corresponds to a function $[\![b]\!] : S \to \mathbb{B}$, defined recursively by $[\![b_1 \wedge b_2]\!](s) = [\![b_1]\!](s) \wedge [\![b_2]\!](s)$ and $[\![b_1 \vee b_2]\!](s) = [\![b_1]\!](s) \vee [\![b_2]\!](s)$. Finally, the syntax of our specifications is given by [5]

$$\phi \;::=\; \texttt{achieve } b \mid \phi_1 \texttt{ ensuring } b \mid \phi_1 ; \phi_2 \mid \phi_1 \texttt{ or } \phi_2,$$

where $b \in \mathcal{P}$. Intuitively, the first construct means that the robot should try to reach a state $s$ such that $[\![b]\!](s) = \texttt{true}$. The second construct says that the robot should try to satisfy $\phi_1$ while always staying in states $s$ such that $[\![b]\!](s) = \texttt{true}$. The third construct says the robot should try to satisfy task $\phi_1$ and then task $\phi_2$. The fourth construct means that the robot should try to satisfy either task $\phi_1$ or task $\phi_2$. Formally, we associate a function $[\![\phi]\!] : Z \to \mathbb{B}$ with $\phi$ recursively as follows:

$$
\begin{aligned}
[\![\texttt{achieve } b]\!](\zeta) &= \exists\, i < t,\; [\![b]\!](s_i) \\
[\![\phi \texttt{ ensuring } b]\!](\zeta) &= [\![\phi]\!](\zeta) \;\wedge\; (\forall i < t,\; [\![b]\!](s_i)) \\
[\![\phi_1 ; \phi_2]\!](\zeta) &= \exists\, i < t,\; ([\![\phi_1]\!](\zeta_{0:i}) \;\wedge\; [\![\phi_2]\!](\zeta_{i:t})) \\
[\![\phi_1 \texttt{ or } \phi_2]\!](\zeta) &= [\![\phi_1]\!](\zeta) \;\vee\; [\![\phi_2]\!](\zeta),
\end{aligned}
$$

where $t$ is the length of $\zeta$. A rollout $\zeta$ *satisfies* $\phi$ if $[\![\phi]\!](\zeta) = \texttt{true}$, which is denoted $\zeta \models \phi$.

**Problem formulation.** Given an MDP and a specification $\phi$, our goal is to compute

$$\pi^* \in \arg\max_\pi \Pr_{\zeta \sim \mathcal{D}_\pi} [[\![\phi]\!](\zeta) = \texttt{true}], \tag{2}$$

where $\mathcal{D}_\pi$ is the distribution over rollouts generated by $\pi$. In other words, we want to learn a policy $\pi^*$ that maximizes the probability that a generated rollout $\zeta$ satisfies $\phi$.

## 3 Compilation and Learning Algorithms

In this section, we describe our algorithm for reducing the above problem (2) for a given MDP $(S, D, A, P, T)$ and a specification $\phi$ to an RL problem specified as an MDP with a reward function. At a high level, our algorithm extends the state space $S$ to keep track of completed subtasks and constructs a reward function $R : Z \to \mathbb{R}$ encoding $\phi$. A key feature of our algorithm is that the user has control over the compilation process—we provide a natural default compilation strategy, but the user can extend or modify our approach to improve the performance of the RL algorithm. We give proofs in Appendix B.

**Quantitative semantics.** So far, we have associated specifications $\phi$ with Boolean semantics (i.e., $[\![\phi]\!](\zeta) \in \mathbb{B}$). A naïve strategy is to assign rewards to rollouts based on whether they satisfy $\phi$:

$$R(\zeta) = \begin{cases} 1 & \text{if } \zeta \models \phi \\ 0 & \text{otherwise.} \end{cases}$$

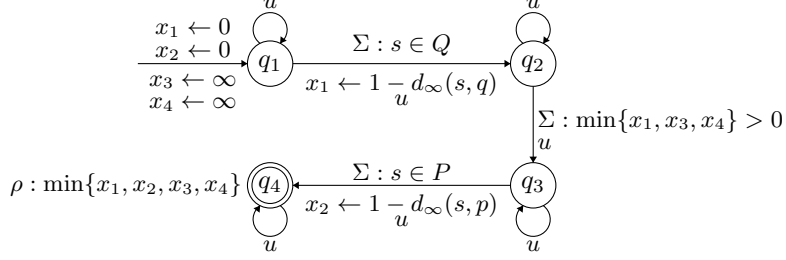

Figure 2: An example of a task monitor. States are labeled with rewards (prefixed with "$\rho$ :"). Transitions are labeled with transition conditions (prefixed with "$\Sigma$ :"), as well as register update rules. A transition from $q_2$ to $q_4$ is omitted for clarity. Also, $u$ denotes the two updates $x_3 \leftarrow \min\{x_3, d_\infty(s, O)\}$ and $x_4 \leftarrow \min\{x_4, \texttt{fuel}(s)\}$.

However, it is usually difficult to learn a policy to maximize this reward due to its discrete nature. A common strategy is to provide a *shaped reward* that quantifies the "degree" to which $\zeta$ satisfies $\phi$. Our algorithm uses an approach based on *quantitative semantics* for temporal logic [7, 8, 14]. In particular, we associate an alternate interpretation of a specification $\phi$ as a real-valued function $[\![\phi]\!]_q : Z \rightarrow \mathbb{R}$. To do so, the user provides quantitative semantics for atomic predicates $p \in \mathcal{P}_0$—in particular, they provide a function $[\![p]\!]_q : S \rightarrow \mathbb{R}$ that quantifies the degree to which $p$ holds for $s \in S$. For example, we can use

$$[\![\texttt{reach } x]\!]_q(s) \;=\; 1 - d_\infty(s, x)$$
$$[\![\texttt{avoid } O]\!]_q(s) \;=\; d_\infty(s, O),$$

where $d_\infty$ is the $L_\infty$ distance between points, with the usual extension to sets. These semantics should satisfy $[\![p]\!]_q(s) > 0$ if and only if $[\![p]\!](s) = \texttt{true}$, and a larger value of $[\![p]\!]_q$ should correspond to an increase in the "degree" to which $p$ holds. Then, the quantitative semantics for predicates $b \in \mathcal{P}$ are $[\![b_1 \wedge b_2]\!]_q(s) = \min\{[\![b_1]\!]_q(s), [\![b_2]\!]_q(s)\}$ and $[\![b_1 \vee b_2]\!]_q(s) = \max\{[\![b_1]\!]_q(s), [\![b_2]\!]_q(s)\}$. Assuming $[\![p]\!]_q$ satisfies the above properties, then $[\![b]\!]_q > 0$ if and only if $[\![b]\!] = \texttt{true}$.

In principle, we could now define quantitative semantics for specifications $\phi$:

$$[\![\texttt{achieve } b]\!]_q(\zeta) = \max_{i < t} [\![b]\!]_q(s_i)$$

$$[\![\phi \texttt{ ensuring } b]\!]_q(\zeta) = \min\{[\![\phi]\!]_q(\zeta), [\![b]\!]_q(s_0), ..., [\![b]\!]_q(s_{t-1})\}$$

$$[\![\phi_1; \phi_2]\!]_q(\zeta) = \max_{i < t} \min\{[\![\phi_1]\!]_q(\zeta_{0:i}), [\![\phi_2]\!]_q(\zeta_{i:t})\}$$

$$[\![\phi_1 \texttt{ or } \phi_2]\!]_q(\zeta) = \max\{[\![\phi_1]\!]_q(\zeta), [\![\phi_2]\!]_q(\zeta)\}.$$

Then, it is easy to show that $[\![\phi]\!](\zeta) = \texttt{true}$ if and only if $[\![\phi]\!]_q(\zeta) > 0$, so we could define a reward function $R(\zeta) = [\![\phi]\!]_q(\zeta)$. However, one of our key goals is to extend the state space so the policy knows which subtasks have been completed. On the other hand, the semantics $[\![\phi]\!]_q$ quantify over all possible ways that subtasks could have been completed in hindsight (i.e., once the entire trajectory is known). For example, there may be multiple points in a trajectory when a subtask $\texttt{reach } q$ could be considered as completed. Below, we describe our construction of the reward function, which is based on $[\![\phi]\!]_q$, but applied to a single choice of time steps on which each subtask is completed.

**Task monitor.** Intuitively, a *task monitor* is a finite-state automaton (FSA) that keeps track of which subtasks have been completed and which constraints are still satisfied. Unlike an FSA, its transitions may depend on the state $s \in S$ of a given MDP. Also, since we are using quantitative semantics, the task monitor has to keep track of the degree to which subtasks are completed and the degree to which constraints are satisfied; thus, it includes *registers* that keep track of the these values. A key challenge is that the task monitor is nondeterministic; as we describe below, we let the policy resolve the nondeterminism, which corresponds to choosing which subtask to complete on each step.

Formally, a task monitor is a tuple $M = (Q, X, \Sigma, U, \Delta, q_0, v_0, F, \rho)$. First, $Q$ is a finite set of *monitor states*, which are used to keep track of which subtasks have been completed. Also, $X$ is a finite set of registers, which are variables used to keep track of the degree to which the specification

holds so far. Given an MDP $(S, D, A, P, T)$, an *augmented state* is a tuple $(s, q, v) \in S \times Q \times V$, where $V = \mathbb{R}^X$—i.e., an MDP state $s \in S$, a monitor state $q \in Q$, and a vector $v \in V$ encoding the value of each register in the task monitor. An augmented state is analogous to a state of an FSA.

The transitions $\Delta$ of the task monitor depend on the augmented state; thus, they need to specify two pieces of information: (i) conditions on the MDP states and registers for the transition to be enabled, and (ii) how the registers are updated. To handle (i), we consider a set $\Sigma$ of predicates over $S \times V$, and to handle (ii), we consider a set $U$ of functions $u : S \times V \to V$. Then, $\Delta \subseteq Q \times \Sigma \times U \times Q$ is a finite set of (nondeterministic) transitions, where $(q, \sigma, u, q') \in \Delta$ encodes *augmented transitions* $(s, q, v) \xrightarrow{a} (s', q', u(s, v))$, where $s \xrightarrow{a} s'$ is an MDP transition, which can be taken as long as $\sigma(s, v) = \mathtt{true}$. Finally, $v_0 \in \mathbb{R}^X$ is the vector of initial register values, $F \subseteq Q$ is a set of final monitor states, and $\rho$ is a reward function $\rho : S \times F \times V \to \mathbb{R}$.

Given an MDP $(S, D, A, P, T)$ and a specification $\phi$, our algorithm constructs a task monitor $M_\phi = (Q, X, \Sigma, U, \Delta, q_0, v_0, F, \rho)$ whose states and registers keep track which subtasks of $\phi$ have been completed. Our task monitor construction algorithm is analogous to compiling a regular expression to an FSA. More specifically, it is analogous to algorithms for compiling temporal logic formulas to automata [23]. We detail this algorithm in Appendix A. The underlying graph of a task monitor constructed from any given specification is acyclic (ignoring self loops) and final states correspond to sink vertices with no outgoing edges (except a self loop).

As an example, the task monitor for $\phi_{\mathrm{ex}}$ is shown in Figure 2. It has monitor states $Q = \{q_1, q_2, q_3, q_4\}$ and registers $X = \{x_1, x_2, x_3, x_4\}$. The monitor states encode when the robot (i) has not yet reached $q$ ($q_1$), (ii) has reached $q$, but has not yet returned to $p$ ($q_2$ and $q_3$), and (iii) has returned to $p$ ($q_4$); $q_3$ is an intermediate monitor state used to ensure that the constraints are satisfied before continuing. Register $x_1$ records $[\![\mathtt{reach}\ q]\!](s) = 1 - d_\infty(s, q)$ when transitioning from $q_1$ to $q_2$, and $x_2$ records $[\![\mathtt{reach}\ p]\!]_q = 1 - d_\infty(s, p)$ when transitioning from $q_3$ to $q_4$. Register $x_3$ keeps track of the minimum value of $[\![\mathtt{avoid}\ s]\!] = d_\infty(s, O)$ over states $s$ in the rollout, and $x_4$ keeps track of the minimum value of $[\![\mathtt{fuel} > 0]\!](s)$ over states $s$ in the rollout.

**Augmented MDP.** Given an MDP, a specification $\phi$, and its task monitor $M_\phi$, our algorithm constructs an *augmented MDP*, which is an MDP with a reward function $(\tilde{S}, \tilde{s}_0, \tilde{A}, \tilde{P}, \tilde{R}, T)$. Intuitively, if $\tilde{\pi}^*$ is a good policy (one that achieves a high expected reward) for the augmented MDP, then rollouts generated using $\tilde{\pi}^*$ should satisfy $\phi$ with high probability.

In particular, we have $\tilde{S} = S \times Q \times V$ and $\tilde{s}_0 = (s_0, q_0, v_0)$. The transitions $\tilde{P}$ are based on $P$ and $\Delta$. However, the task monitor transitions $\Delta$ may be nonderministic. To resolve this nondeterminism, we require that the policy decides which task monitor transitions to take. In particular, we extend the actions $\tilde{A} = A \times A_\phi$ to include a component $A_\phi = \Delta$ indicating which one to take at each step. An *augmented action* $(a, \delta) \in \tilde{A}$, where $\delta = (q, \sigma, u, q')$, is only available in augmented state $\tilde{s} = (s, q, v)$ if $\sigma(s, v) = \mathtt{true}$. Then, the *augmented transition probability* is given by,

$$\tilde{P}((s, q, v),\ (a, (q, \sigma, u, q'))),\ (s', q', u(s, v))) = P(s, a, s').$$

Next, an *augmented rollout* of length $t$ is a sequence $\tilde{\zeta} = (s_0, q_0, v_0) \xrightarrow{a_0} \dots \xrightarrow{a_{t-1}} (s_t, q_t, v_t)$ of augmented transitions. The *projection* $\mathrm{proj}(\tilde{\zeta}) = s_0 \xrightarrow{a_0} \dots \xrightarrow{a_{t-1}} s_t$ of $\tilde{\zeta}$ is the corresponding (normal) rollout. Then, the *augmented rewards*

$$\tilde{R}(\tilde{\zeta}) = \begin{cases} \rho(s_T, q_T, v_T) & \text{if } q_T \in F \\ -\infty & \text{otherwise} \end{cases}$$

are constructed based on $F$ and $\rho$. The augmented rewards satisfy the following property.

**Theorem 3.1.** *For any MDP, specification $\phi$, and rollout $\zeta$ of the MDP, $\zeta$ satisfies $\phi$ if and only if there exists an augmented rollout $\tilde{\zeta}$ such that (i) $R(\tilde{\zeta}) > 0$, and (ii) $\mathrm{proj}(\tilde{\zeta}) = \zeta$.*

Thus, if we use RL to learn an optimal *augmented policy* $\tilde{\pi}^*$ over augmented states, then $\tilde{\pi}^*$ is more likely to generate rollouts $\tilde{\zeta}$ such that $\mathrm{proj}(\tilde{\zeta})$ satisfies $\phi$.

**Reward shaping.** As discussed before, our algorithm constructs a shaped reward function that provides "partial credit" based on the degree to which $\phi$ is satisfied. We have already described one step of reward shaping—i.e., using quantitative semantics instead of the Boolean semantics.

However, the augmented rewards $\tilde{R}$ are $-\infty$ unless a run reaches a final state of the task monitor. Thus, our algorithm performs an additional step of reward shaping—in particular, it constructs a reward function $\tilde{R}_s$ that gives partial credit for accomplishing subtasks in the MDP.

For a non-final monitor state $q$, let $\alpha : S \times Q \times V \to \mathbb{R}$ be defined by

$$\alpha(s, q, v) = \max_{(q, \sigma, u, q') \in \Delta, \ q' \neq q} [\![\sigma]\!]_q(s, v).$$

Intuitively, $\alpha$ quantifies how "close" an augmented state $\tilde{s} = (s, q, v)$ is to transitioning to another augmented state with a different monitor state. Then, our algorithm assigns partial credit to augmented states where $\alpha$ is larger.

However, to ensure that a good policy according to the shaped rewards $\tilde{R}_s$ is also a good policy according to $\tilde{R}$, it does so in a way that preserves the ordering of the cumulative rewards for rollouts— i.e., for two length $T$ rollouts $\tilde{\zeta}$ and $\tilde{\zeta}'$, it guarantees that if $\tilde{R}(\tilde{\zeta}) > \tilde{R}(\tilde{\zeta}')$, then $\tilde{R}_s(\tilde{\zeta}) > \tilde{R}_s(\tilde{\zeta}')$.

To this end, we assume that we are given a lower bound $C_\ell$ on the final reward achieved when reaching a final monitor state—i.e., $C_\ell < \tilde{R}(\tilde{\zeta})$ for all $\tilde{\zeta}$ with final state $\tilde{s}_T = (s_T, q_T, v_T)$ such that $q_T \in F$ is a final monitor state. Furthermore, we assume that we are given an upper bound $C_u$ on the absolute value of $\alpha$ over non-final monitor states—i.e., $C_u \geq |\alpha(s, q, v)|$ for any augmented state such that $q \notin F$.

Now, for any $q \in Q$, let $d_q$ be the length of the longest path from $q_0$ to $q$ in the graph of $M_\phi$ (ignoring self loops in $\Delta$) and $D = \max_{q \in Q} d_q$. Given an augmented rollout $\tilde{\zeta}$, let $\tilde{s}_i = (s_i, q_i, v_i)$ be the first augmented state in $\tilde{\zeta}$ such that $q_i = q_{i+1} = ... = q_T$. Then, the shaped reward is

$$\tilde{R}_s(\tilde{\zeta}) = \begin{cases} \max_{i \leq j < T} \alpha(s_j, q_T, v_j) + 2C_u \cdot (d_{q_T} - D) + C_\ell & \text{if } q_T \notin F \\ \tilde{R}(\tilde{\zeta}) & \text{otherwise.} \end{cases}$$

If $q_T \notin F$, then the first term of $\tilde{R}_s(\tilde{\zeta})$ computes how close $\tilde{\zeta}$ was to transitioning to a new monitor state. The second term ensures that moving closer to a final state always increases reward. Finally, the last term ensures that rewards $\tilde{R}(\tilde{\zeta})$ for $q_T \in F$ are always higher than rewards for $q_T \notin F$. The following theorem follows straightforwardly.

**Theorem 3.2.** *For two augmented rollouts $\tilde{\zeta}, \tilde{\zeta}'$, (i) if $\tilde{R}(\tilde{\zeta}) > \tilde{R}(\tilde{\zeta}')$, then $\tilde{R}_s(\tilde{\zeta}) > \tilde{R}_s(\tilde{\zeta}')$, and (ii) if $\tilde{\zeta}$ and $\tilde{\zeta}'$ end in distinct non-final monitor states $q_T$ and $q_T'$ such that $d_{q_T} > d_{q_T'}$, then $\tilde{R}_s(\tilde{\zeta}) \geq \tilde{R}_s(\tilde{\zeta}')$.*

**Reinforcement learning.** Once our algorithm has constructed an augmented MDP, it can use any RL algorithm to learn an *augmented policy* $\tilde{\pi} : \tilde{S} \to \tilde{A}$ for the augmented MDP:

$$\tilde{\pi}^* \in \arg\max_{\tilde{\pi}} \mathbb{E}_{\tilde{\zeta} \sim \mathcal{D}_{\tilde{\pi}}}[\tilde{R}_s(\tilde{\zeta})]$$

where $\mathcal{D}_{\tilde{\pi}}$ denotes the distribution over augmented rollouts generated by policy $\tilde{\pi}$. We solve this RL problem using augmented random search (ARS), a state-of-the-art RL algorithm [15].

After computing $\tilde{\pi}^*$, we can convert $\tilde{\pi}^*$ to a *projected policy* $\pi^* = \text{proj}(\tilde{\pi}^*)$ for the original MDP by integrating $\tilde{\pi}^*$ with the task monitor $M_\phi$, which keeps track of the information needed for $\tilde{\pi}^*$ to make decisions. More precisely, $\text{proj}(\tilde{\pi}^*)$ includes internal memory that keeps track of the current monitor state and register value $(q_t, v_t) \in Q \times V$. It initializes this memory to the initial monitor state $q_0$ and initial register valuation $v_0$. Given an augmented action $(a, (q, \sigma, u, q')) = \tilde{\pi}^*((s_t, q_t, v_t))$, it updates this internal memory using the rules $q_{t+1} = q'$ and $v_{t+1} = u(s_t, v_t)$.

Finally, we use a neural network architecture similar to neural module networks [3, 2], where different neural networks accomplish different subtasks in $\phi$. In particular, an augmented policy $\tilde{\pi}$ is a set of neural networks $\{N_q \mid q \in Q\}$, where $Q$ are the monitor states in $M_\phi$. Each $N_q$ takes as input $(s, v) \in S \times V$ and outputs an augmented action $N_q(s, v) = (a, a') \in \mathbb{R}^{k+2}$, where $k$ is the out-degree of the $q$ in $M_\phi$; then, $\tilde{\pi}(s, q, v) = N_q(s, v)$.

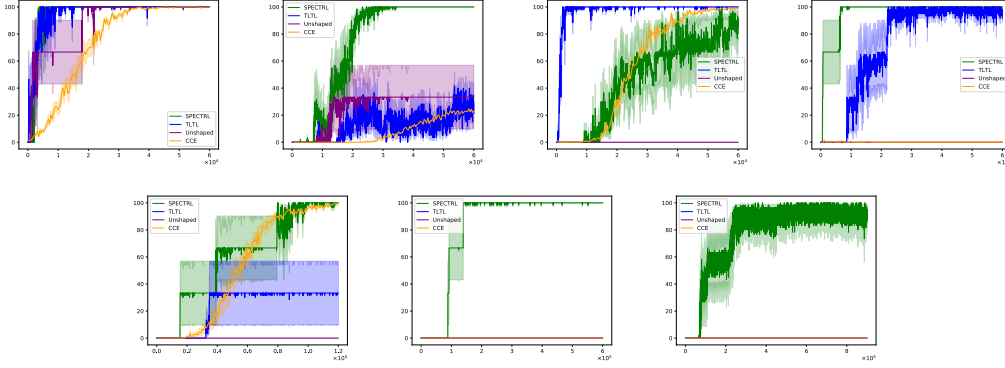

Figure 3: Learning curves for $\phi_1$, $\phi_2$, $\phi_3$ and $\phi_4$ (top, left to right), and $\phi_5$, $\phi_6$ and $\phi_7$ (bottom, left to right), for SPECTRL (green), TLTL (blue), CCE (yellow), and SPECTRL without reward shaping (purple). The $x$-axis shows the number of sample trajectories, and the $y$-axs shows the probability of satisfying the specification (estimated using samples). To exclude outliers, we omitted one best and one worst run out of the 5 runs. The plots are the average over the remaining 3 runs with error bars indicating one standard deviation around the average.

## 4  Experiments

**Setup.** We implemented our algorithm in a tool SPECTRL[6], and used it to learn policies for a variety of specifications. We consider a dynamical system with states $S = \mathbb{R}^2 \times \mathbb{R}$, where $(x, r) \in S$ encodes the robot position $x$ and its remaining fuel $r$, actions $A = [-1, 1]^2$ where an action $a \in A$ is the robot velocity, and transitions $f(x, r, a) = (x + a + \epsilon, r - 0.1 \cdot |x_1| \cdot \|a\|)$, where $\epsilon \sim \mathcal{N}(0, \sigma^2 I)$ and the fuel consumed is proportional to the product of speed and distance from the $y$-axis. The initial state is $s_0 = (5, 0, 7)$, and the horizon is $T = 40$.

In Figure 3, we consider the following specifications, where $O = [4, 6] \times [4, 6]$:

- $\phi_1 = $ achieve reach $(5, 10)$ ensuring $($avoid $O)$
- $\phi_2 = $ achieve reach $(5, 10)$ ensuring $($avoid $O \wedge (r > 0))$
- $\phi_3 = $ achieve $($reach $[(5, 10); (5, 0)])$ ensuring avoid $O$
- $\phi_4 = $ achieve $($reach $(5, 10)$ or reach $(10, 0)$; reach $(10, 10))$ ensuring avoid $O$
- $\phi_5 = $ achieve $($reach $[(5, 10); (5, 0); (10, 0)])$ ensuring avoid $O$
- $\phi_6 = $ achieve $($reach $[(5, 10); (5, 0); (10, 0); (10, 10)])$ ensuring avoid $O$
- $\phi_7 = $ achieve $($reach $[(5, 10); (5, 0); (10, 0); (10, 10); (0, 0)])$ ensuring avoid $O$

where the abbreviation achieve $(b; b')$ denotes achieve $b$; achieve $b'$ and the abbreviation reach $[p_1; p_2]$ denotes reach $p_1$; reach $p_2$. For all specifications, each $N_q$ has two fully connected hidden layers with 30 neurons each and ReLU activations, and tanh function as its output layer. We compare our algorithm to [13] (TLTL), which directly uses the quantitative semantics of the specification as the reward function (with ARS as the learning algorithm), and to the constrained cross entropy method (CCE) [24], which is a state-of-the-art RL algorithm for learning policies to perform tasks with constraints. We used neural networks with two hidden layers and 50 neurons per layer for both the baselines.

**Results.** Figure 3 shows learning curves of SPECTRL (our tool), TLTL, and CCE. In addition, it shows SPECTRL without reward shaping (Unshaped), which uses rewards $\tilde{R}$ instead of $\tilde{R}_s$. These plots demonstrate the ability of SPECTRL to outperform state-of-the-art baselines. For specifications $\phi_1, ..., \phi_5$, the curve for SPECTRL gets close to 100% in all executions, and for $\phi_6$ and $\phi_7$, it gets close to 100% in 4 out of 5 executions. The performance of CCE drops when multiple constraints (here, obstacle and fuel) are added (i.e., $\phi_2$). TLTL performs similar to SPECTRL on tasks $\phi_1$, $\phi_3$ and $\phi_4$ (at least in some executions), but SPECTRL converges faster for $\phi_1$ and $\phi_4$.

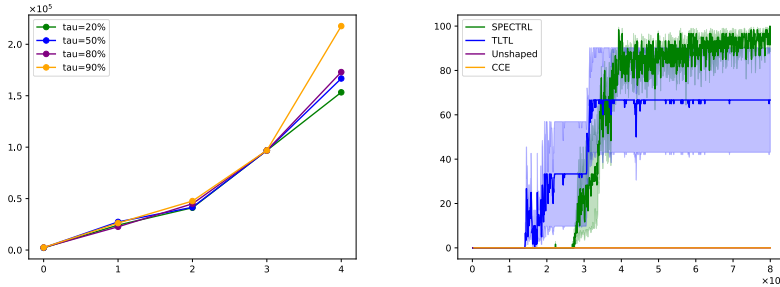

Figure 4: Sample complexity curves (left) with number of nested sequencing operators on the x-axis and average number of samples to converge on the y-axis. Learning curve for cartpole example (right).

Since TLTL and CCE use a single neural network to encode the policy as a function of state, they perform poorly in tasks that require memory—i.e., $\phi_5$, $\phi_6$, and $\phi_7$. For example, to satisfy $\phi_5$, the action that should be taken at $s = (5, 0)$ depends on whether $(5, 10)$ has been visited. In contrast, SPECTRL performs well on these tasks since its policy is based on the monitor state.

These results also demonstrate the importance of reward shaping. Without it, ARS cannot learn unless it randomly samples a policy that reaches final monitor state. Reward shaping is especially important for specifications that include many sequencing operators $(\phi; \phi')$—i.e., specifications $\phi_5$, $\phi_6$, and $\phi_7$.

Figure 4 (left) shows how sample complexity grows with the number of nested sequencing operators $(\phi_1, \phi_3, \phi_5, \phi_6, \phi_7)$. Each curve indicates the average number of samples needed to learn a policy that achieves a satisfaction probability $\geq \tau$. SPECTRL scales well with the size of the specification.

**Cartpole.** Finally, we applied SPECTRL to a different control task—namely, to learn a policy for the version of cart-pole in OpenAI Gym, in which we used continuous actions instead of discrete actions. The specification is to move the cart to the right and move back left without letting the pole fall. The formal specification is given by

$$\phi = \texttt{achieve}\ (\texttt{reach}\ 0.5; \texttt{reach}\ 0.0)\ \texttt{ensuring}\ \texttt{balance}$$

where the predicate `balance` holds when the vertical angle of the pole is smaller than $\pi/15$ in absolute value. Figure 4 (right) shows the learning curve for this task averaged over 3 runs of the algorithm along with the three baselines. TLTL is able to learn a policy to perform this task, but it converges slower than SPECTRL; CCE is unable to learn a policy satisfying this specification.

## 5  Conclusion

We have proposed a language for formally specifying control tasks and an algorithm to learn policies to perform tasks specified in the language. Our algorithm first constructs a task monitor from the given specification, and then uses the task monitor to assign shaped rewards to runs of the system. Furthermore, the monitor state is also given as input to the controller, which enables our algorithm to learn policies for non-Markovian specifications. Finally, we implemented our approach in a tool called SPECTRL, which enables the users to *program* what the agent needs to do at a high level; then, it automatically learns a policy that tries to best satisfy the user intent. We also demonstrate that SPECTRL can be used to learn policies for complex specifications, and that it can outperform state-of-the-art baselines.

**Acknowledgements.** We thank the reviewers for their insightful comments. This work was partially supported by NSF by grant CCF 1723567 and by AFRL and DARPA under Contract No. FA8750-18-C-0090.

## Footnotes

[1]Intuitively, this construction is analogous to compiling a regular expression to a finite state automaton.

[2]SPECTRL stands for SPECifying Tasks for Reinforcement Learning.

[3]Note that we consider rollout-based rewards rather than state-based rewards. Most modern RL algorithms, such as policy gradient algorithms, can use rollout-based rewards.

[4]Formally, a predicate is a string in the context-free language generated by this context-free grammar.

[5]Here, $\texttt{achieve}$ and $\texttt{ensuring}$ correspond to the "eventually" and "always" operators in temporal logic.

[6]The implementation can be found at `https://github.com/keyshor/spectrl_tool`.

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
