[Supplementary Material · spectrl_supplement.pdf]

# A    Task Monitor Construction Algorithm

We describe our algorithm for constructing a task monitor $M_\phi$ for a given specification $\phi$. Our construction algorithm proceeds recursively on the structure of $\phi$. Implicitly, our algorithm maintains the property that every monitor-state $q$ has a self-transition $(q, \mathtt{true}, u, q)$; here, the update function $u$ is the identity by default, but may be modified as part of the construction.

**Notation.** We use $\sqcup$ to denote the disjoint union. Given $v \in \mathbb{R}^X$, $v' \in \mathbb{R}^{X'}$, we define $v'' = v \oplus v' \in \mathbb{R}^{X \sqcup X'}$ to be their concatenation—i.e.,

$$v''(x) = \begin{cases} v(x) & \text{if } x \in X \\ v'(x) & \text{otherwise.} \end{cases}$$

Given $v \in \mathbb{R}^X$ and $Y \subseteq X$, we define $v \downarrow_Y \in \mathbb{R}^Y$ to be the restriction of $v$ to $Y$. Given $v \in \mathbb{R}^X$ and $Y \supseteq X$, we define $v' = \mathtt{extend}(v)_Y \in \mathbb{R}^Y$ to be

$$v'(y) = \begin{cases} v(y) & \text{if } y \in X \\ 0 & \text{otherwise.} \end{cases}$$

We drop the subscript $Y$ when it is clear from context. Finally, given $v \in \mathbb{R}^X$ and $k \in \mathbb{R}$, we define $v' = v[x \mapsto k] \in V$ to be

$$v(x') = \begin{cases} v(x') & \text{if } x' \neq x \\ k & \text{otherwise.} \end{cases}$$

Also, recall that a predicate $b \in \mathcal{P}$ is defined over states $s \in S$; it can straightforwardly be extended to a predicate in $\Sigma$ over $(s, v) \in S \times V$ by ignoring $v$. Note that every predicate $b \in \mathcal{P}$ is a negation free Boolean combination of atomic predicates $p \in \mathcal{P}_0$.

Finally, the definition of $\Sigma$ depends on the set $X$ of registers in the task monitor. When necessary, we use the notation $\Sigma_X$ to make this dependence explicit. Finally, for $X \subseteq X'$, any $\sigma \in \Sigma_X$ can be interpreted as a predicate in $\Sigma_{X'}$ by ignoring the components of $X'$ not in $X$.

**Objectives.** Consider the case $\phi = \mathtt{achieve}\ b$, where $b \in \mathcal{P}$. For this specification, our algorithm constructs the following task monitor:

The initial state is marked with an arrow into the state. Final states are double circles. Predicates $\sigma \in \Sigma$ labeling a transition appear prefixed by "$\Sigma$ :". Rewards $\rho$ labeling a state appear prefixed by "$\rho$ :". Self loops are associated with the true predicate (omitted). Updates $u \in U$ are by default the identity function. Intuitively, the state $q_1$ on the left encodes when subtask $b$ is not yet completed, and the state $q_2$ on the right encodes when $b$ is completed, and $x_1$ records the degree to which $b$ is satisfied upon completion.

**Constraints.** Consider the case $\phi = \phi_1\ \mathtt{ensuring}\ b$, where $b \in \mathcal{P}$. Let

$$M_{\phi_1} = (Q_1, X_1, \Sigma, U_1, \Delta_1, q_1^0, v_1^0, F_1, \rho_1)$$

Then, $M_\phi$ is the product of $M_\phi$ and $M_{\mathtt{ensuring}\ b}$ defined as

$$M_\phi = (Q_1, X_1 \sqcup \{x_b\}, \Sigma, U, \Delta, q_1^0, v_0, F_1, \rho),$$

where $M_{\mathtt{ensuring}\ b}$ is

Figure 5: Overview of monitor construction for sequencing and choice operators.

Then, we have $(q, \sigma, u', q') \in \Delta$ if and only if there is a transition $(q, \sigma, u, q') \in \Delta_1$ such that

$$u'(s, v) = \texttt{extend}(u(s, v \downarrow_{X_1}))[x_b \mapsto \min(v(x_b), [\![b]\!](s))].$$

Furthermore, the initial register valuation $v_0 = \texttt{extend}(v_1^0)[x_b \mapsto \infty]$ and the reward function $\rho$ is

$$\rho(s, q, v) = \min\{\rho_1(s, q, v \downarrow_{X_1}), v(x_b)\}.$$

Intuitively, $x_b$ encodes the minimum degree to which $b$ is satisfied during a rollout.

**Sequencing.** Third, consider $\phi = \phi_1; \phi_2$. Intuitively, $M_\phi$ is constructed by concatenating the registers of $M_{\phi_1}$ and $M_{\phi_2}$ (extending the update functions $u$ as needed), and adding transitions $(q, \sigma, u, q_0)$ from each final state $q$ of $M_{\phi_1}$ to the initial state $q_0$ of $M_{\phi_2}$, where $\sigma = \texttt{true}$ and $u$ is the identity on registers for $M_{\phi_1}$ and sets the registers of $M_{\phi_2}$ to their initial values. A subtle issue is that transitioning from $\phi_1$ to $\phi_2$ takes one time step, yet it should take zero time steps. Therefore, we add transitions from each final state of $M_{\phi_1}$ to all successors of the initial state of $M_{\phi_2}$.

More precisely, let

$$M_{\phi_1} = (Q_1, X_1, \Sigma, U_1, \Delta_1, q_1^0, v_1^0, F_1, \rho_1)$$
$$M_{\phi_2} = (Q_2, X_2, \Sigma, U_2, \Delta_2, q_2^0, v_2^0, F_2, \rho_2).$$

Assume without loss of generality that $X_2 \subseteq X_1$. Then,

$$M_\phi = (Q_1 \sqcup Q_2, X_1 \sqcup \{x_R\}, \Sigma, U, \Delta, q_1^0, v_0, F_2, \rho).$$

Here, $\Delta = \Delta_1' \cup \Delta_2' \cup \Delta_{1 \to 2}$, where $(q, \sigma, u', q') \in \Delta_i'$ if there exists $(q, \sigma, u, q') \in \Delta_i$ such that

$$u'(s, v) = u(s, v \downarrow_{X_i}) \oplus v \downarrow_{X \setminus X_i},$$

and $(q, \sigma' \wedge \sigma_R, u', q') \in \Delta_{1 \to 2}$ if $q \in F_1$ and there exists $(q_2^0, \sigma, u, q') \in \Delta_2$ such that the atomic predicate $\sigma_R$ is given by

$$[\![\sigma_R]\!](s, v) = \rho_1(s, q, v \downarrow_{X_1}) > 0,$$

the predicate $\sigma'$ is given by

$$[\![\sigma']\!](s, v) = [\![\sigma]\!](s, v_2^0),$$

and $u'$ is given by

$$u'(s, v) = \mathtt{extend}(u(s, v_2^0))[x_R \mapsto \rho_1(s, q, v \downarrow_{X_1})].$$

The initial register valuation is $v_0 = \mathtt{extend}(v_1^0)$, and the reward function $\rho$ is

$$\rho(s, q, v) = \min\{\rho_2(s, q, v \downarrow_{X_2}), v(x_R)\}$$

for all $q \in F_2$.

**Choice.** Consider the case $\phi = \phi_1$ or $\phi_2$. Intuitively, $M_\phi$ is constructed by combining the initial states of $M_{\phi_1}$ and $M_{\phi_2}$ into a single initial state $q_0$, and concatenating their registers. The transitions from $q_0$ are the union of the transitions from the initial states of $M_{\phi_1}$ and $M_{\phi_2}$. More precisely, let

$$M_{\phi_1} = (Q_1, X_1, \Sigma, U_1, \Delta_1, q_1^0, v_1^0, F_1, \rho_1)$$
$$M_{\phi_2} = (Q_2, X_2, \Sigma, U_2, \Delta_2, q_2^0, v_2^0, F_2, \rho_2).$$

This construction assumes that there are self loops on the initial states of $M_{\phi_1}$ and $M_{\phi_2}$. Then,

$$M_\phi = (Q, X_1 \sqcup X_2, \Sigma, U, \Delta, q_0, v_1^0 \oplus v_2^0, F_1 \sqcup F_2, \rho).$$

Here,

$$Q = (Q_1 \setminus \{q_1^0\}) \sqcup (Q_2 \setminus \{q_2^0\}) \sqcup \{q_0\},$$

and $\Delta = \Delta_1' \cup \Delta_2' \cup \Delta_0$, where where $(q, \sigma, u', q') \in \Delta_i'$ if $q \neq q_0$ and there is a transition $(q, \sigma, u, q') \in \Delta_i$ such that

$$u'(s, r, v) = \mathtt{extend}(u(s, r, v \downarrow_{X_i})).$$

Also, let $(q_1^0, \top, u_1^0, q_1^0) \in \Delta_1$ and $(q_2^0, \top, u_2^0, q_2^0) \in \Delta_2$ be the self loops on the initial states of $M_{\phi_1}$ and $M_{\phi_2}$ respectively. Let

$$u_0(s, r, v) = u_1^0(s, r, v \downarrow_{X_1}) \oplus u_2^0(s, r, v \downarrow_{X_2}).$$

Then, $(q_0, \sigma, u', q) \in \Delta_0$ if either (i) $(q_0, \sigma, u', q) = (q_0, \top, u_0, q_0)$, or (ii) there exists $i \in \{1, 2\}$ such that $(q_i^0, \sigma, u, q) \in \Delta_i$, where $q \in Q_i \setminus \{q_i^0\}$ and

$$u'(s, r, v) = \mathtt{extend}(u(s, r, v \downarrow_{X_i})).$$

The reward function $\rho$ for $q \in F_i$ is given by:

$$\rho(s, q, v) = \rho_i(s, q, v \downarrow_{X_i}).$$

# B  Proofs of Theorems

## B.1  Proof of Theorem 3.1

First, the following lemma follows by structural induction:

**Lemma B.1.** *For $\sigma \in \Sigma$, $[\![\sigma]\!](s, v) = \mathtt{true}$ if and only if $[\![\sigma]\!]_q(s, v) > 0$.*

Next, let $G_M$ denote the underlying state transition graph of a task monitor $M$. Then,

**Lemma B.2.** *The task monitors constructed by our algorithm satisfy the following properties:*

1. *The only cycles in $G_M$ are self loops.*

2. *The finals states are precisely those states from which there are no outgoing edges except for self loops in $G_M$.*

3. *In $G_M$, every state is reachable from the initial state and for every state there is a final state that is reachable from it.*

4. *For any pair of states $q$ and $q'$, there is at most one transition from $q$ to $q'$.*

5. *There is a self loop on every state $q$ given by a transition $(q, \top, u, q)$ for some update function $u$ where $\top$ denotes the true predicate.*

The first three properties ensure progress when switching from one monitor state to another. The last two properties enable simpler composition of task monitors. The proof follows by structural induction. Theorem 3.1 now follows by structural induction on $\phi$ and Lemmas B.1 and B.2.

## B.2 Proof of Theorem 3.2

(i) Let $\tilde{\zeta}, \tilde{\zeta}'$ be two augmented rollouts such that $\tilde{R}(\tilde{\zeta}) > \tilde{R}(\tilde{\zeta}')$. There are three cases to consider:

- Case A. Both $\tilde{\zeta}$ and $\tilde{\zeta}'$ end in final monitor states: In this case, $\tilde{R}_s(\tilde{\zeta}) = \tilde{R}(\tilde{\zeta}) > \tilde{R}(\tilde{\zeta}') = \tilde{R}_s(\tilde{\zeta}')$ as required.

- Case B. $\tilde{\zeta}$ ends in a final monitor state but $\tilde{\zeta}'$ ends in a monitor state $q_T' \notin F$: In this case,

$$
\begin{aligned}
\tilde{R}_s(\tilde{\zeta}') &= \max_{i' \leq j < T} \alpha(s_j', q_T', v_j') + 2C_u \cdot (d_{q_T} - D) + C_\ell \\
&\leq \max_{i' \leq j < T} \alpha(s_j', q_T', v_j') - 2C_u + C_\ell && (d_{q_T} \leq D - 1) \\
&\leq C_\ell && (C_u \text{ is an upper bound on } \alpha) \\
&< \tilde{R}(\tilde{\zeta}) && (C_\ell \text{ is a lower bound on } \tilde{R}) \\
&= \tilde{R}_s(\tilde{\zeta}).
\end{aligned}
$$

- Case C. $\tilde{\zeta}$ ends in non-final monitor state: In this case $\tilde{R}(\tilde{\zeta}) = -\infty$ and hence the claim is vacuously true.

(ii) Let $\tilde{\zeta}, \tilde{\zeta}'$ be two augmented rollouts ending in distinct (non-final) monitor states $q_T$ and $q_T'$ such that $d_{q_T} > d_{q_T'}$. Then,

$$
\begin{aligned}
\tilde{R}_s(\tilde{\zeta}) &= \max_{i \leq j < T} \alpha(s_j, q_T, v_j) + 2C_u \cdot (d_{q_T} - D) + C_\ell \\
&\geq \max_{i \leq j < T} \alpha(s_j, q_T, v_j) + 2C_u + 2C_u \cdot (d_{q_T'} - D) + C_\ell && (d_{q_T} \geq d_{q_T'} - 1 \ \& \ C_u \geq 0) \\
&\geq C_u + 2C_u \cdot (d_{q_T'} - D) + C_\ell && (C_u \text{ is an upper bound on } -\alpha) \\
&\geq \max_{i' \leq j < T} \alpha(s_j', q_T', v_j') + 2C_u \cdot (d_{q_T'} - D) + C_\ell && (C_u \text{ is an upper bound on } \alpha) \\
&= \tilde{R}_s(\tilde{\zeta}').
\end{aligned}
$$