[Reviews · NeurIPS 2019]

Reviewer 1



The paper proposes a language for expressing task specifications for reinforcement learning (RL). The language is based on linear temporal logic and is compiled into a finite-state automaton (FSA). An augmented MDP that contains the FSA’s states as a part of its state is then constructed and equipped with a shaped reward function that rewards the agent for being close towards completing subtasks without violating constraints. The paper is clearly written. It is very heavy on notation, but this is probably unavoidable in such a paper. All notation is clearly explained, and with a bit of effort, the method can be understood. Empirical evaluation is limited, but it does show big improvements over carefully chosen baselines. As a non-expert in the topic of task specifications for RL, the reviewer can’t 100% guarantee that the method is novel, but the related work section seems rather compelling. In particular, it seems that the construction of the augmented MDP and reward shaping indeed make the paper sufficiently novel for publication. A question: using augmented MDP effectively introduces a sort of memory. Could not the same be achieved via using something like an LSTM? Another question: it is not very clear from the paper what the SPECTRL system is, and it probably should not be mentioned in the bullet list of contributions in the introduction. UPD: I have read the rebuttal and other reviews, I still think the paper is good. I would highly recommend the authors to release a reference implementation.

Reviewer 2



Significance: I'm not sure Spectrl enables RL to tackle new problems it couldn't before, given that the experiments are on relatively simple tasks and don't have a few baselines I'd like to see (see the "Improvements" section of my review). The approach also seems pretty specific to a certain class of problems (ones that can be specified in the particular formal language). That's fine as long as it results in significant improvements on that class of problems or enables tackling new problems entirely; I wasn't sure that was the case given the existing experiments, at least enough to warrant it's use over simpler RL methods that don't require the propose framework and its added implementation overhead / reward shaping. Quality: I'd have liked to see more complex/challenging experiments that validate Spectrl enables something new to be done, or else that it's widely applicable or applicable to harder tasks than grid worlds. Prior work e.g. on TLTL shows experiments with an actual robotic arm, for instance. I'm also not sure how to interpret the existing experiments, given there aren't error bars and a few baselines I'd find helpful (see "Improvements" section of my review). Clarity: Overall, the writing was clear. The method itself took a while to explain (and a lot of math), so it may help to shorten that. The conclusion is missing, though, and explanation of the final experiment (cart pole) and its results are cut short (just one paragraph). Originality: Spectrl seems somewhat close to TLTL, except for the task monitor which can help handle non-Markovian specifications. However, I would have liked to see more experiments (and on more complex environments) that highlight that Spectrl can handle non-Markovian specifications specifically. I'm not too familiar with TLTL, but it sounds like it would be a small modification to enable TLTL to handle non-Markovian cases as well (i.e., add some extra history to the state, though if I'm wrong, please let me know in the rebuttal).

Reviewer 3



The specification language seems to be similar to past work, being a restricted form of temporal logic. The atomic predicates comes in two flavours: (“eventually”) achieve certain state or (“always”) ensuring to avoid certain states. Various composition of these atomic predicates can be used (A then B, A or B, etc.). The paper’s proposed finite state machine “task monitor” bears resemblance to the FSM “reward machines” proposed by Icarte et al. [1], which was not cited/discussed. So I will be quite interested how the authours clarify its differences to the Reward Machines. It appears that the main contribution is in the use of reward shaping and theoretical analysis for the shaped reward consistency. On the quantitative semantics: an example of using L_infinity distance metric was given to quantify the degree at which the agent achieves the subtask. I am wondering how this type of reward shaping affects the SPECTRL agent when there are say physical barriers in the environment, which leads to some local minimas if the agent relies on the L_infinity metric for the reaching subtask? The experiment in cartpole OpenAI gym appears incomplete -- is it possible to compare SPECTRL to the other baselines as in the other environment here? I found the paper to be fairly difficult to follow due to the large number of notations introduced. It may be helpful to include: A high level diagram used to convert from the task specification language -> Task Monitor -> Augmented MDP. A concise algorithm box for SPECTRL. I am aware that there is a detailed algorithm description in the appendix, but perhaps a brief minimal summary of the SPECTRL framework in this form will make it quite clear to the reader. For the plots, I suggest having a more standard plotting of average of the 5 runs (as done) and also include a shaded area with standard deviation over the seeds so that we can understand the stability of SPECTRL. Each of the axis should also labeled (in addition to the current caption description). *** Post Author Response *** I think that the authors did a reasonable job in their response contrasting Reward Machines (RM) to SPECTRL. They clarified that (1) in RM, the FSM is directly given to the agent, while here the contribution is in the conversion from formal language to the task monitor, and (2) The task monitor also contains registers, making it more powerful than FSM. On (1), I do want to point out a concurrent work by Camacho et al [2] to appear in IJCAI 2019 which follows up on RM by using formal language to specify reward and automatically construct the RM and use reward shaping (potential-based as in Ng et al 1999). I only found out about the work now and hence did not mention it in my original review, so I am not trying to penalize the authors for not mentioning/comparing to this work. Rather, I think this concurrent work at least supports that it's a research direction that people in the community are pursuing and SPECTRL will be of interest. 2. On environments, I agree that their existing environment contains obstacles anyways. The clarification on the cartpole environment was also important, as I'm sure that they ran out of space in the paper to properly discuss (at least could have included these modification details in the appendix). But other reviewers have said, we still don't have baselines for the cartpole environment so that will be helpful to add / discuss. Overall I am increasing my score from my initial review. Reference: [1] Icarte, R.T., Klassen, T., Valenzano, R. & McIlraith, S.. (2018). Using Reward Machines for High-Level Task Specification and Decomposition in Reinforcement Learning. Proceedings of the 35th International Conference on Machine Learning, in PMLR 80:2107-2116 [2] LTL and Beyond: Formal Languages for Reward Function Specification in Reinforcement Learning. Camacho, A.; Icarte, R. T.; Klassen, T. Q.; Valenzano, R.; and McIlraith, S. A. In Proceedings of the Twenty-Eighth International Joint Conference on Artificial Intelligence (IJCAI), 2019. To appear. (PDF: http://www.cs.toronto.edu/~toryn/papers/IJCAI-2019.pdf)

[Author Response · NeurIPS 2019]

We thank the reviewers for their helpful suggestions, and will do our best to incorporate them into our paper. Overall,
we want to emphasize that the goal of SPECTRL is to make it easier to apply RL to tasks with complex objectives. In
particular, SPECTRL enables the user *program* what the agent needs to do at a high level; then, SPECTRL automatically
learns a policy that tries to best satisfy the user intent.

**R1, LSTMs:** It is true that LSTMs can be used to solve RL problems with non-Markovian specifications. However,
the task monitor introduced in this paper not only eliminates the need to learn the internal state, but also enables us to
perform reward shaping. As we show in our experiments, reward shaping is crucial for learning complex tasks. Thus,
even if the LSTM learns a perfect encoding of the state, our approach would still substantially outperform it.

**R2, Curiosity-driven exploration:** These approaches aren't really applicable to addressing our challenges, which
involve complex objectives rather than hard exploration. For example, the task may be for the agent to visit a sequence
of goals in a particular order. The agent may know how to reach each of these goals individually, but this is not sufficient
for the agent to complete the task.

**R2, Sub-goal based rewards.** Many challenges arise when considering the details of how sub-goal based rewards
would be implemented. For example, how does achieving a sub-goal count compared to violating a constraint? How do
we handle sub-goals that can be achieved in multiple ways? How do we ensure the agent does not repeatedly obtain a
reward for a sub-goal that it has already completed? As tasks get more complex/deeply nested, manually determining
rewards for sub-goals is very non-trivial. These challenges are exactly what our system is designed to solve.

**R2, Baselines:** We used ARS since it has been demonstrated to be state-of-the-art for continuous control. For fair
comparison, we use ARS with TLTL as well. Also, like TLTL, our approach can be used with with other algorithms.

**R2, More challenging environments:** Our experiments already show that our algorithm can solve problems that the
state-of-the-art ARS algorithm cannot. Benchmarks such as MuJoCo may involve more complex dynamics, but they
focus largely on short-term control tasks—e.g., running in a straight line as fast as possible. In contrast, the benefit of
our approach is on handling complex, long-term control tasks, where the agent must perform a variety of actions to
achieve complex objectives—e.g., reaching multiple goals in sequence and then coming back. While there has been
some success solving these problems using traditional RL, these approaches typically rely on enormous amounts of
computation. In contrast, our algorithm converges quickly without a large amount of computation (all our experiments
are run on a single GPU).

**R2, Cart-Pole:** Our cart-pole benchmark differs from the standard OpenAI one in two ways. (i) The goal in the
standard one is just to keep the pole balanced for as long as possible. In ours, the goal is to move the cart to the right
and then back to the starting position without letting the pole fall. We find this task to be substantially more challenging
than the standard one. (ii) The standard benchmark has discrete actions, whereas ours has continuous actions, which
also makes the task more challenging.

**R2, TLTL:** We have two contributions over TLTL. (i) While it is possible to extend the state space to handle non-
Markovian TLTL specifications, this could not previously be done automatically. One of our contributions is to provide
a way for automatically extending the state space; our algorithm can be modified to work for TLTL formulas as well.
(ii) We perform reward shaping, whereas TLTL does not. We find that reward shaping is critical for good performance.

**R3, Reward machines:** We differ from the paper on reward machines in two ways. First, in their paper, the specification
is given directly as a finite state machine along with reward functions for each state. Their key contribution is in the
ability to learn multiple tasks simultaneously by applying the Q-learning updates across different specifications. In
contrast, our contribution is to automatically generate a task monitor from a high-level logical specification. Second,
our notion of task monitor is more expressive since it has a finite set of registers that can store real values. In contrast,
finite state machines cannot store quantitative information. We will add a discussion to our related work.

**R3, barriers:** Some of our benchmarks include barriers (i.e., obstacles). Our approach is able to solve these tasks.

**Presentation:** To improve readability, we plan to add pictures describing the high level approach in a final version. We
will include error-bar plots for Figure 3, which we show below. To exclude outliers, we omitted one best and one worst
run out of the 5 runs. While the variance is sometimes high, our insights continue to hold—in particular, our algorithm
consistently outperforms the baseliness. We will add a conclusion as well.



[Meta-Review · NeurIPS 2019]

The paper presents and evaluates SPECTRL, a framework for transforming formal specifications of tasks into shaped reward function. The reviewers agreed that, while it is not obvious that this paper will be extremely impactful, it is nonetheless interesting, convincing, and clearly written. After some discussion, the consensus leans towards acceptance, although with some outstanding issues (especially regarding the cartpole results) which should be addressed before publication. It is also highly recommended that a reference implementation of this method be released for use within the community, although it is not in my power to make this a formal requirement for publication.